# Influence of Litter Size at Birth on Productive Parameters in Guinea Pigs (*Cavia porcellus*)

**DOI:** 10.3390/ani10112059

**Published:** 2020-11-07

**Authors:** Angela Edith Guerrero Pincay, Raúl Lorenzo González Marcillo, Walter Efraín Castro Guamàn, Nelson Rene Ortiz Naveda, Deyvis Angel Grefa Reascos, Santiago Alexander Guamàn Rivera

**Affiliations:** 1Escuela Superior Politècnica de Chimborazo, Sede Orellana, El Coca 220150, Ecuador; raul.gonzales@espoch.edu.ec (R.L.G.M.); efrain.castro@espoch.edu.ec (W.E.C.G.); nelson.ortiz@espoch.edu.ec (N.R.O.N.); 2Ministerio de Agricultura y Ganaderìa, Proyecto Nacional de Innovaciòn Tecnològica Participativa y Productividad Agrìcola (PITPPA), El Coca 220150, Ecuador; dgrefar@mag.gob.ec; 3Ruminant Research Group (G2R), Animal and Food Science Department Universitat Autònoma de Barcelona, 08193 Bellatera, Spain; santiagoalexander.guaman@e-campus.uab.cat

**Keywords:** yield, carcass, feed conversion, breeding

## Abstract

**Simple Summary:**

*Cavia porcellus* is an autochthonous species of the high Andean region of countries such as Ecuador, Bolivia, Colombia, and Peru. Several works have been conducted to create programs aimed at improving their productive characteristics. Thus, breeding and conservation are very important for an animal species of high precocity and prolificacy, providing a source of protein of high biological value and for being part of the cultural legacy of many of these countries. Currently, this species is part of the food security in high Andean areas, and technology is being incorporated into its production at an industrial level, also establishing marketing channels. Herein, we conclude that the best productive responses regarding weight were in the litters of three guinea pigs. Furthermore, with respect to sex, the males presented better productive performance than the females.

**Abstract:**

A study was conducted at the Escuela Superior Politècnica de Chimborazo, Ecuador, to evaluate the influence of litter size of guinea pigs (*Cavia porcellus*) on their development and to establish the economic profitability of the production system. Forty-eight animals were used, distributed into litters of two, three, and four rodents per litter, with a balanced diet and green fresh alfalfa for the weaning, growth, and fattening stage, the rodents and litters were randomly selected, applying the statistical model completely randomly and evaluating different variables across 120 days. The litters of three guinea pigs obtained the best productive responses and economic profitability. With respect to sex, the males presented better productive behavior, greater economic increase, and less cost, evidencing that mixed feeding influences the number of guinea pigs per birth in terms of growth and development. The results serve to improve guinea pig meat production for the rural population.

## 1. Introduction

The guinea pig or cavy (*Cavia* spp.) was domesticated for food purposes in the highlands of Peru, Ecuador, Colombia, and Bolivia at least 7000 years ago and its descendants are still widely used as a source of meat throughout South America. Nowadays, it is a small domestic mammal, *Cavia porcellus*, that stands out for its precocity, prolificacy, and diet flexibility, which, together with the excellent quality of its meat being similar in appearance to rabbits or chickens, makes it a source of protein capable of competing with other domestic species of productive interest [1]. This rodent has a great adaptability to a wide range of housing and management options, being widely used by rural smallholders. Moreover, this small rodent is identified within the life and customs of the indigenous society, it is also used in medicine and even in magical–religious rituals [2]. Additionally, this species has numerous litters and a high reproductive rate [3].

Guinea pigs are multiple-ovulation animals, so they can have between one to six offspring per birth, with an average of two, occasionally presenting up to eight per litter [4]. The weight at birth and individual weaning are characteristics of economic importance that have positive correlations with the final weight [5]. In guinea pigs of the Peruvian breed, a weight of 752.4 ± 126.1 g at nine weeks, a carcass weight of 420 ± 54 g [6], and a carcass yield of between 62.76% and 69.87% can be reached [7].

One of the reasons for the interest in studying guinea pig exploitation is the need for meat production from a herbivorous species that easily adapts to different ecosystems and is fed with fresh green fodder [7]. The commercial exploitation of guinea pigs is a way to reach new management alternatives, tending to achieve better use of food resources. Diverse institutions have introduced guinea pig breeds and genetic lines to the market in order to produce meat, and these animals have been crossed with native guinea pigs, thereby improving the yield of carcasses, food conversion, and the income of breeders [6].

Livestock plays an important role in the lives of humans as converters, recyclers, and banks of nutrients. Smallholders raise a diversity of livestock species [8]. One of the main economic alternatives in the rural sector of Riobamba in Chimborazo, Ecuador, is guinea pig production, which has become a feeding option in the Andean region. Due to the demand for guinea pig meat, the producers seek to optimize breeding and management techniques [9]. Sustainable farming is a technique that seeks to achieve economic and social profitability without harming the environment and that is important for identifying sustainability through the efficient use of biological resources. Moreover, it maintains a balance with the environment in which it takes place [10].

The productive yields of guinea pigs under intensive conditions are similar to domestic animals due to the advances in genetics, management, and installations, which influence the productive and reproductive parameters. This investigation was conducted in the small species production unit of the School of Livestock Sciences, Riobamba, Ecuador. Our aim was to determine the influence of litter size on the productive parameters in guinea pigs *(Cavia porcellus*). Therefore, the present study evaluated the productive parameters to determine the best performance in this species, as well as its profitability. These findings could then be disseminated to the rural population of Riobamba, Ecuador, due to their economic, social, and ancestral importance.

## 2. Materials and Methods

This study was carried out at the Productive Unit of Minor Species, Facultad de Ciencias Pecuarias of the Escuela Superior Politécnica del Chimborazo, Riobamba, Ecuador, located between the coordinates 78°26′ W (longitude) and 1°25′ S (latitude) at 2740 m above sea level (masl).

In Ecuador, commercial exploitation has not been able to reach a technical level, among the influencing factors are defective sanitary management, inadequate breeds for the environment, adaptation of exogenous (*Peruvian*) technologies that are not adequate, and, above all, the most preponderant factor is food [9]

### 2.1. Animals, Treatments, and Handling Conditions

Fifty-six primiparous female type 1 Peruvian guinea pigs aged between 3 and 4 months with a body weight of 0.900 ± 0.50 kg underwent reproduction with a total of 7 fertile male type 1 Peruvian breeding guinea pigs (4 months old with a body weight of 1.1 ± 0.70 kg selected for their high heritability for meat production) in one relation (10:1). We ensured that the animals met the additional criterion of not showing evidence of systemic or physical disorders. After parturition from a population of 102 guinea pigs, 48 young animals (24 males and 24 females) from litters of different sizes at birth (2, 3, and 4 rodents per litter) were taken and distributed into 24 pens, where the pen was the experimental unit and was formed by two animals of the same sex. The present investigation did not experience any mortality, possibly because hygiene and health precautions were taken during the study.

All animal care, housing, and feeding procedures were adapted based on the World Organisation for Animal Health 2016 (animal welfare) and the current Ecuadorian regulations: Organic Law on Animal Health No. 56, published in the Official Gazette, Supplement 27, 03 July 2017. In addition to the guide for the care and use of laboratory animals, the Division of Land and Life Studies 2001 was adhered to.

The food was given according to the age of the guinea pigs. With a balanced diet (i.e., green alfalfa (mixed food)), they were given green alfalfa (100 g) plus a concentrate (50% ground corn and 13.7% wheat bran and 7% rice powder and 28% dehydrated alfalfa and 1% salt and 0.3% antioxidant). Increasing the amounts as the age progressed, consumption was measured every 24 h based on the daily surplus and weight measurements that were taken before providing the food.

During lactation and weaning (15 days of age), the guinea pigs were fed a balanced diet, which contained 18% gross protein, digestible energy of 3000 kcal/kg food, 8% gross fiber, 1.3% calcium, 0.6% phosphorus, and 50 g of green alfalfa. In the growth and fattening stage (105 days old), they were fed a balanced diet that contained 17% gross protein, digestible energy of 2963 kcal/kg food, 15% crude fiber, 0.13% calcium, 1.03% phosphorus, and 100 g of fresh green alfalfa.

### 2.2. Feed Consumption

Feed was supplied according to the age of the animals, starting at the weaning stage with 10 g of feed and 100 g of fresh alfalfa per animal, to gradually increase these amounts as the age progressed until reaching the consumption of 25 g of concentrate and 175 g of forage and 80 mL of water in the growth and fattening stage. Consumption was measured every 24 h based on the daily surplus, and weights were recorded using a 5 kg scale

### 2.3. Sample Collection, Analyses, and Measurements: Weights at Birth, Weaning, and End

Body weight was measured at birth, weaning (15 days of age), and at the end of the study (105 days of age), ensuring that the animals were weighed before being fed. The weight was taken through a balance, capacity 2000 g ± 0.5 precision (GRAM FC, Madrid, Spain). The guinea pigs were weighed at 7:00 a.m. on an empty stomach, before the day’s food was given. The control of the weights of the guinea pigs in each of the experimental units was done at the beginning of the fieldwork (from birth), then at 15 days (weaning), and then at the end of the experiment (105 days old). To determine the weight gain, the initial and final weights of the guinea pigs were considered.

The measurements of weight increase and weaning were calculated by the difference between the final and initial weight of each evaluation phase. Feed conversion was determined based on the total feed consumption in dry matter divided by weight gain. In addition, the weight of the carcass was determined after being sacrificed, considering a clean carcass in which the head was included, but not the blood, hair, or viscera. Moreover, for the carcass yield, 50% of the animals were slaughtered.

### 2.4. Economic Analysis

The cost per kilogram of weight gain was established based on the costs of the feed consumed (forage and balance) multiplied by the feed conversion.

### 2.5. Statistical Analyses

The experimental data were processed with the statistical package SPSS (IBM SPSS Statistics, H. Nie, C. Hadlai) using analyses of variance and separation of means according to Tukey’s test at the significance level of *p <* 0.05 and tendency was considered *p* < 0.10 unless otherwise indicated. In addition, the trend lines were determined through polynomial regression in the variables that registered differences because of the evaluated litter size.

The variables evaluated were weight at birth, weaning (15 days of age), and end of study (105 days), weight gain at weaning (up to 15 days), total weight gain (at 105 days), feed consumption in kilograms of dry matter (DM), forage consumption in kilograms of DM, total feed consumption in kilograms of DM, feed conversion, cost per kilogram of weight gain (USD), carcass weight in kilograms, carcass yield %, and profit/cost.

The experimental units were under a completely random design in a combinatorial arrangement (factor A = litter size; factor B = the sex of the animals in the case of the offspring). For the litters, a completely random design was used in a monofactorial arrangement (litter size), with 4 repetitions per treatment and an experimental unit of 2 animals adjusted to the following additive models.

For evaluation of the offspring by effect of the litter size at birth (factor A) and the sex of the animals (factor B), it was:Y_ijk_ = µ + A_i_ + B_j_ +AB_ij_ + E_ijk_(1)
where µ is the general average; A_i_ is the effect of litter size at birth (2, 3, and 4 offspring per litter); B_j_ is the effect of animal sex; AB_ij_ refers to the interaction A × B; E_ijk_ is the residual error.

Meanwhile, for evaluating the litter according to size at birth, it was the following:Yij = µ + Ti + Eij(2)
where µ is the general average; Ti is the effect of litter size at birth (2, 3, and 4 offspring per litter); Eij is the effect of the experimental error.

## 3. Results

### 3.1. Weight at Birth

Tendency (*p* < 0.10) was detected in the individual average weight at birth between the litters of two, three, and four guinea pigs per calving. Guinea pigs corresponding to three rodents per calving had a better individual weight than that of the two and three rodents per calving. In terms of animal sex, no statistically significant differences were found (*p* > 0.05), but the males had higher numerical weight at birth than the females. Moreover, the interaction A × B did not present differences (*p* < 0.05) in any of the studied parameters.

### 3.2. Weaning Weight at 15 Days

No differences were found (*p* < 0.05) in the weaning weight at 15 days between litters, although the animals of two guinea pigs per calving had higher weight. Additionally, in terms of the effect of sex, both (males and females) had similar weaning weight at 15 days (*p* > 0.05) (Table 1). Thus, it can be deduced that during lactation, male and female guinea pigs had the same opportunities to take advantage of the food and to present equal body development.

### 3.3. Weight at 105 Days

The effect of litter size at birth resulted in differences in weight at 105 days (*p* = 0.003). In this sense, the highest weight was found in the litters of three guinea pigs per birth, followed by two guinea pigs per birth, while the lowest weight was found for the litters of four guinea pigs per birth. According to sex, the male guinea pigs were heavier than the females; nevertheless, only a trend was detected (*p* < 0.10; Table 1).

Therefore, through regression analysis, a quadratic tendency of 26.32% was established, which highlights that the weight of the animals increased when the size of the litter was greater than two guinea pigs per birth, but decreased when the size of the litter was greater than three guinea pigs per birth (Figure 1).

In addition, a linear trend regression of 87.16% was also determined, which indicates that the greater the number of animals per litter, the greater the total weight at weaning (Figure 2).

Additionally, with the regression analysis, a quadratic trend of 91.08% was determined, establishing that the greater the number of animals per litter, the greater the weight of the litter at the end of the fattening, although the behavior was not proportional (Figure 3).

### 3.4. Weaning Gain Weight at 15 Days

In terms of gain weight, there were no significant differences *(p* > 0.32), although the animals of two rodents per litter presented a superior weight. According to sex, the increase in weight was not significantly different *(p* < 0.05), although the females obtained a greater increase in weight than the males. Figure 4 indicates that the weight increases were a function of the size of the litter, although the behavior was not proportional, but always in an upward direction.

### 3.5. Weight Total 105 Days

The greatest increases in total weight (at 105 days of age) were registered in the litter of three animals per birth (*p* < 0.05), followed two animals per birth, and finally, the litter of four guinea pigs per birth presented a lower weight gain (Table 1). According to the sex of the animals, only a trend was observed (*p <* 0.10), i.e., the males had a better weight than the females (0.602 vs. 0.548 kg, respectively). Additionally, the regression analysis determined a quadratic trend of 90.14%, from which it follows that the weight gain tended to increase in an unequal way according to the size of the litter.

### 3.6. Carcass Weight

According to the productive behavior by litter size at birth, the carcass weight was higher for the animals from litters of three guinea pigs per birth (*p <* 0.05), compared with the weights of the guinea pig carcasses of four and two guinea pigs per litter. In the same way, the effect of sex had no significant difference (*p =* 0.225), but the weight of the male carcasses was greater than that of the females.

The regression analysis determined a quadratic tendency of 24.70%, showing that the weight of the carcasses was greater from animals of three guinea pigs per litter, but it was reduced when the number of guinea pigs per litter was greater. Therefore, statistically, these responses confirm that the sex of the animals does not influence productive responses, even though better behavior was determined in the males.

In terms of productive behavior by litter, the total weight of the carcasses presented differences at *p* < 0.05, since the highest total weights were determined in the litters of four animals per birth. A carcass weight of 1.75 ± 0.2 kg was obtained for the litters of four animals per litter, followed by the carcasses with three animals per litter (1.54 ± 0.2 kg), while the litters of two animals per litter obtained weights of 0.99 ± 0.2 kg. Through regression analysis, a quadratic tendency of 90.6% was determined, indicating that the greater the number of animals per litter, the greater the weight of the carcasses of the litter, although the behavior was not proportional (Figure 5).

### 3.7. Carcass Yield

The analysis of the means according to Tukey’s test did not present significant differences (at *p* < 0.01) by the effect of litter size at birth. Similarly, in terms of sex, the yields found were statistically similar (Table 1).

### 3.8. Feed Consumption

With respect to the productive behavior by breeding, the analysis of means, according to Tukey’s test in terms of the consumption of a balanced diet by breeding, was different (*p* < 0.05), since it was established that greater consumption was achieved by the litter of three animals per litter (2.17 ± 0.2 kg), being reduced to 2.11 ± 0.2 kg in the two animals per litter and to 1.90 ± 0.2 kg in the four animals per litter groups. These findings suggest that the balanced consumption increased when the litter size was three animals per birth but tended to decrease when the number of animals per litter at birth was greater. The sex of the animals did not influence the consumption (*p* > 0.05), even though the males consumed a greater amount of feed than the females.

The amount of forage consumed during the growth and fattening stage was not statistically different, since alfalfa forage was provided in fixed amounts, reaching a consumption of all forage provided, so that all animals, both males and females, consumed 1.05 ± 0.1 kg of forage in dry matter. A significant difference of *p* < 0.05 was found for the total consumption of food by litter size, with the highest consumption (3.22 kg ± 0.1) being obtained for the guinea pigs from three animals per litter.

Differences of *p* < 0.01 were also found due to the consumption of food having a direct relation with the number of animals per litter and with the increases in weight, the greater the corporal development and the greater the consumption of food. The amount of forage consumed varied (*p* < 0.01), and it was found that litters with four animals per birth consumed 4.20 ± 0.6 kg, followed by litters with three animals per birth that consumed 3.15 ± 0.6 kg of forage, while litters with two animals per birth consumed 2.10 ± 0.6 kg. In relation to the total feed consumption, significant differences at *p* < 0.01 were found in the averages analyzed according to Tukey’s test, maintaining that the most numerous litters (four animals per litter) registered the highest feed consumption (11.79 kg).

Moreover, the regression analysis determined a quadratic tendency of 97.63%, which indicates that feed consumption was a function of the number of animals per litter, because the greater the number of animals per litter, the greater the feed consumption to cover its nutritional requirements.

### 3.9. Food Conversion

According to the productive behavior by litter, the food conversion of the guinea pigs was not significantly different (*p <* 0.05). It was determined that the animals required between 5.32 ± 0.3 and 5.71 ± 0.3 kg of food for each kilogram of weight gain, corresponding to the animals from the three and four animals per litter groups, respectively. According to sex, the food conversions were 5.57 ± 0.2 kg and 5.35 ± 0.2 kg in the female and male animals, respectively. This shows that in spite of a lack of statistical differences in weight and feed consumption, better responses were achieved in the litters from the three animals per birth. The food conversion was similar in the guinea pigs that presented the lowest weights and feed consumption, which shows that, all in all, the food provided covered the nutritional requirements during the development, growth, and fattening of the animals. In relation to the productive behavior by litter, the food conversion averages analyzed with Tukey’s test did not present significant differences (*p <* 0.05), since the determined values varied between 5.30 ± 0.4 and 5.68 ± 0.4 kg, which corresponds to the litters of three and four animals per birth, respectively.

### 3.10. Economic Analysis: Cost Per Kilogram of Weight Gain

In terms of productive behavior by breeding, the cost of production per kilogram of weight gain was not statistically different (*p <* 0.05). However, they registered a lower cost of production of 2.16 ± 0.1 USD/kg of weight gain when they came from the three animals per litter group and 2.30 ± 0.1 USD/kg of weight gain for the litters of four animals per birth. When comparing these values, which are the two extreme cases, an income of 14 cents/kg of weight gain can be established. Likewise, the effect of sex on the cost of production was similar; in males, each kilogram of weight gain costs USD 2.17 ± 0.04, while it costs USD 2.25 ± 0.04 in females, which denotes an income of 8 cents/kg of weight gain. In terms of the productive behavior by litter, the means analysis using Tukey’s test resulted in the cost per kilogram of weight gain, which presented no difference (*p >* 0.05). When the litter size was four animals per birth, the production cost was 2.29 ± 0.04 USD/kg of weight gain, which was reduced to 2.17 ± 0.04 USD/kg of weight gain in litters of two animals per birth and 2.15 ± 0.04 USD/kg of weight gain in litters of three animals per birth.

The economic analysis based on the expenditures and income generated (Table 2) showed that there is a higher profitability when animals from litters of three animals per birth are exploited, because the benefit/cost indicator was USD 1.15. For this litter size, for each dollar invested, a profit of USD 0.15 was obtained, followed by litters of two animals per birth with a profit/cost of USD 1.11, as opposed to litters of four animals per birth with a profit/cost of less than USD 1.04. Therefore, the best productive and economic responses were presented by the litters of three animals per birth. According to the sex of the animals, the highest profitability can be achieved with the breeding of male animals (a profit of USD 0.19 for each dollar invested).

## 4. Discussion

When comparing the results of the productive behavior by animal with the productive behavior by litter, it was determined that they differed in weight at birth, since the productive behavior by animal was the best in terms of weight for the litter of three animals per birth and for the productive behavior by the litter of four guinea pigs per birth. The productive behavior by animal was related to the fact that the larger the litter size, the lower the individual weight of the animals [4].

Therefore, there was an inverse relationship between the size of the litter and the weight of the guinea pigs at birth. The values obtained in the present work are within the range established in the results of the experiment conducted by [11], and are related to the fact that an increase in the number of animals decreases the weight at birth, and that the variation is mainly due to the number of animals obtained at birth.

Global poverty and food insecurity continue to remain critical issues, especially in rural areas [8]. During the Conference on Environment and Development in Rio in 1992, sustainable development was mentioned as a guiding principle for all economic and political sectors [12].

Currently, at the commercial level, the selection of animals of high productive performance with some phenotypic traits according to the demand of the market is very important. Through this study, an optimal litter size of three rodents per calving was established from the point of view of reproduction. These findings contribute to the sustainability of guinea pig meat production at the family and commercial levels, as well as to the mitigation of hunger and poverty in the rural and urban populations of Riobamba, Ecuador.

With respect to sex at weaning, it was determined that during lactation, both males and females have the same opportunities to take advantage of food, which is in agreement with that mentioned by other authors who carried out similar research. When comparing weight at weaning, we found differences in terms of the litters of two animals per birth having better productive behavior by breeding, indicating that the bigger the litter size, the smaller the individual weight of the animals, with the productive behavior of the litters of four animals per birth being the poorest.

The litter factor, as a random effect, as well as birth weight, had a marked influence on the weaning weight of the offspring [5]. In relation to the final weights, the weight in terms of both the productive behavior by guinea pig and by litter was found to be the highest in the litters of three animals per birth, noting a slight increase in the productive behavior by litter in the four animals per birth group. This indicates that the greater the number of animals per litter, the greater the weight of the litter at the end of fattening. In relation to the final weights by litter for productive behavior, the best weights were found in males, which is in agreement with other research indicating that male guinea pigs have better assimilation of nutrients, thereby obtaining better bone structure and strengthening their immune system [9]. The males (*n* = 2609, 147.3 ± 33.0 g) were heavier than the females [5].

With respect to weight gain at weaning, both factors differed, with the productive behavior by animals being higher in terms of weight in the litters of two guinea pigs per birth and the productive behavior by litter having the greatest increases in the larger litters (i.e., four animals per birth). This highlights that weight increases are a function of litter size, which is consistent with the fact that the weight of newborns is largely related to litter size [11].

In relation to the productive behavior of the guinea pigs, in terms of the weight gain according to sex, the males presented better responses than the females, which coincides with the results of another author who showed that the male guinea pigs present better use of the food supplied than the females [13]. In contrast, when comparing the productive behavior by breeding with the productive behavior by litter, it was found that the feed consumption differed. There was a greater intake in the litters of three guinea pigs per birth in terms of productive behavior by breeding, which indicates that balanced feed consumption increases in this litter size and tends to decrease when the number of guinea pigs per birth is greater. This behavior could be due to the body weight of the animals, since the greater the weight, the greater the amount of food the animals require to meet their nutritional requirements, as well as productive behavior.

The highest consumption was obtained by litters with four animals per birth, which is because consumption has a direct relationship with the number of animals per litter. Moreover, their weight increases were higher, since the greater the body development, the greater the feed consumption. The responses related to the productive behavior by litter, leading to variations in the results, were due to the size of the animals.

Since there is an effect associated with competition for food, the smaller the number of animals per litter, the greater the availability of food; the opposite occurs when the litter is larger than three animals per birth. Additionally, in terms of the productive behavior by litter, the guinea pigs in larger litters consume more solid food than the guinea pigs in smaller litters [14]. With regard to food conversion, the productive behavior by breeding and by litter was not different between litters of three and four animals per birth.

The determination of feed consumption should be based on the individual behavior of the animals, as the food provided should cover the nutritional requirements during the development, growth, and fattening of the animals. Related to this, as mentioned by the author of [15], responses are subordinated to the individuality of the animals to take advantage of the food, as well as the type and quality of the food ration provided. Comparing the productive behavior by breeding and by litter with the production cost per kilogram of weight gain, a lower production cost was determined when using animals from the three animals per litter group [16].

The results obtained emphasize the importance of feeding costs in guinea pig meat production [17]. In a farm destined for any animal production, the main objective is to produce a large quantity in a short time and with low production costs that generate profitability. The results of this investigation, in relation to the productive behavior by breeding based on sex, show that male guinea pigs acquire greater development in the growth and fattening stage than females.

With respect to carcass weight, the productive behavior by breeding and by litter differs. In the first case, the highest weights were found in the litters of three guinea pigs per birth, which reduced when the number of guinea pigs increased. In the second case, the highest weights were determined in the litters of four guinea pigs per birth, evidencing that the greater the number of guinea pigs per litter, the greater the weight of the litters, although the behavior is subordinated to the individuality of the animals to take advantage of the food, as well as the type and quality of the food ration provided [13]. Comparing the productive behavior by breeding and by litter with carcass yield, it was determined that the animals from the three guinea pigs per birth group obtained better results, which could be increased by better genetics and pasture forage–concentrate ratio of 70:30. Alternatively, feeding guinea pigs with a balanced ration can improve the carcass yields [13,18].The sex of the animals influences the productive parameters, with males being dedicated to meat production and females to reproduction [13].

## 5. Conclusions

The productive behavior by breeding and by litter showed that the litter size of the guinea pigs influenced the productive responses from birth to fattening, food conversion, cost per kilogram of weight gain, and carcass yield. Consequently, the best productive responses regarding weight were in the litters of three guinea pigs per birth, and during the breeding and fattening of the guinea pigs, the litters of four guinea pigs per birth had the least productive responses. In the productive behavior by breeding and by litter with respect to sex, the male animals presented better productive behavior than the females. Additionally, it was determined that the greatest economic profitability could be achieved when fattening guinea pigs from litters with three offspring per birth, with a cost-benefit of USD 1.15 compared to the USD 1.04 obtained from litters of four offspring per birth.

## Figures and Tables

**Figure 1 animals-10-02059-f001:**
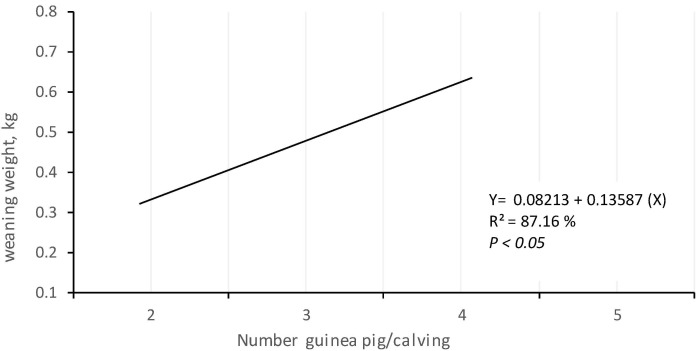
Performance of the final weights (105 days old) of guinea pigs of both sexes due to the effect of litter size at birth.

**Figure 2 animals-10-02059-f002:**
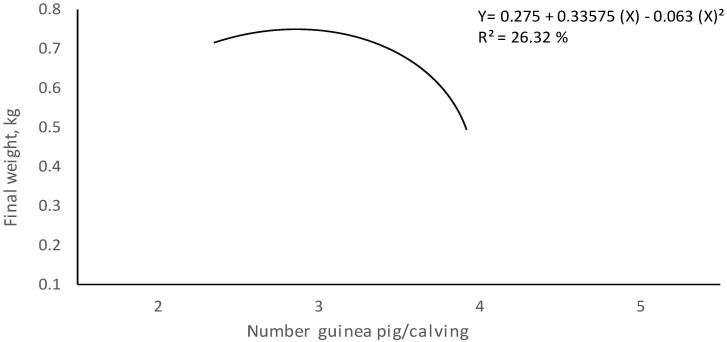
Behavior of the weight of litter (kg) of the guinea pigs according the animals/calving at weaning (15 days).

**Figure 3 animals-10-02059-f003:**
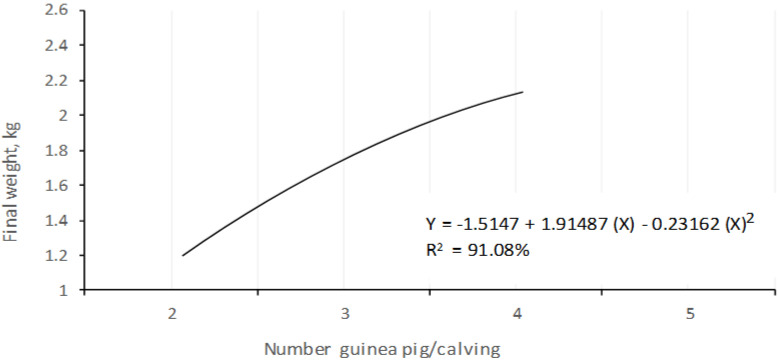
Behavior of the weight of litter (kg) of the guinea pigs with different number animals/calving at 105 days.

**Figure 4 animals-10-02059-f004:**
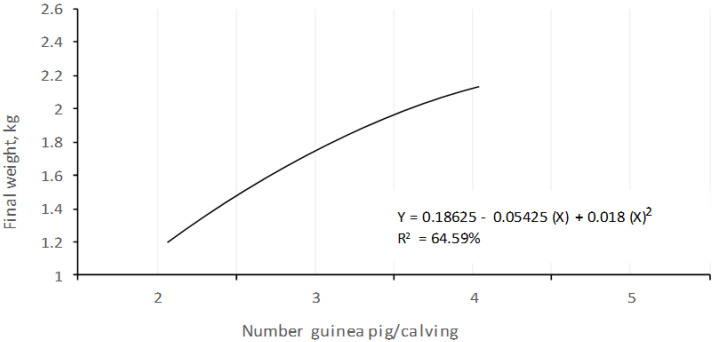
Behavior of the weight gain (g) of the guinea pigs with different number animals/calving at weaning (15 days).

**Figure 5 animals-10-02059-f005:**
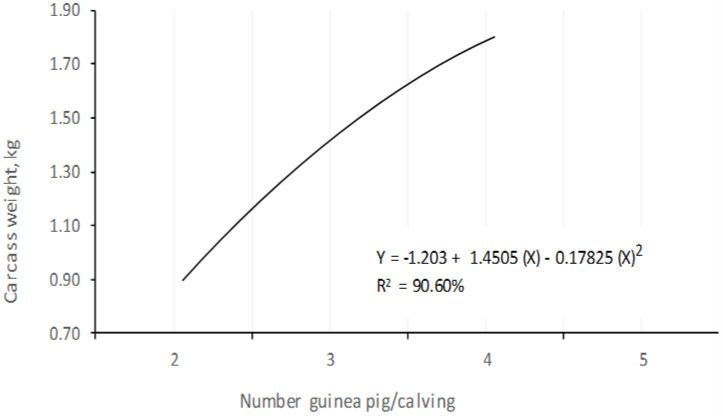
Performance of carcass weight (kg) of guinea pigs with different number animals/calving at 105 days of age.

**Table 1 animals-10-02059-t001:** Productive parameter measurement, with respect to litter size at birth and effect of sex in the guinea pigs.

Item	Litter Size at Birth			Sex		SE	Effect Value ^1^		
	2	3	4	Male	Female		A	B	A × B
Weight birth, kg	0.009	0.105	0.091	0.101	0.096	0.004	0.062	0.265	0.125
Weaning weight 15 days, kg	0.174	0.167	0.155	0.165	0.167	0.007	0.198	0.805	0.425
Weight at 105 days, kg	0.695	0.715	0.610	0.703	0.643	0.023	0.003	0.075	0.245
Weaning gain weight 15 days, kg	0.075	0.062	0.065	0.063	0.071	0.006	0.324	0.314	0.750
Weight total 105 days, kg	0.595	0.610	0.519	0.602	0.548	0.023	0.046	0.082	0.225
Carcass weight, kg	0.493	0.515	0.437	0.503	0.459	0.020	0.040	0.225	0.327
Carcass, yield %	70.9	71.9	71.6	71.5	71.4	0.780	0.722	0.874	0.540

^1^ Effect of litter size (A), and animal sex of guinea pigs (B) as well as the interaction A × B. Statistical differences were declared at *p* < 0.05 and tendency at *p* < 0.10 unless otherwise indicated.

**Table 2 animals-10-02059-t002:** Economic evaluation (USD) of guinea pig production by different litter size (from 1 to 105 days old).

Item	Litter size			Sex	
	2	3	4	Females	Males
Number of animals	16	16	16	24	24
Animal costs ^1^	16	16	16	24	24
Cost of food					
Forage ^2^	3.53	3.53	3.53	5.29	5.29
Balanced ^3^	11.83	12.12	10.62	16.65	17.92
Health ^4^	3.2	3.2	3.2	7.2	7.2
Labour ^5^	40	40	40	60	60
Total expenditures	74.56	74.85	73.35	113.14	114.41
Sale carcass guinea pigs ^6^	63.1	65.92	55.94	88.13	96.58
Subscription sale ^7^	20	20	20	40	40
Total income	83.10	85.92	75.94	128.13	136.58
Benefit/Cost	1.11	1.15	1.04	1.13	1.19

^1^ USD/1.00 each (pups) at birth; ^2^ USD 0.21 each kg of alfalfa in dry matter; ^3^ USD 0.35 each kg of balanced in dry matter; ^4^ USD 0.20 per animal; ^5^ USD 30.00 monthly wage; ^6^ USD 8.00 each kg of carcass; ^7^ USD 20.00 fertilizer.

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
