# Peer review of "Influence of Litter Size at Birth on Productive Parameters in Guinea Pigs (Cavia porcellus)"

_animals, 2020, doi:10.3390/ani10112059_

Round 1
Reviewer 1 Report
The research presents an interesting analysis of the economic profitability of breeding guinea pigs. The results of the above research have a practical dimension and highlight factors that may affect the final, economic result of breeding. Although the nature of the research is not particularly innovative and the number of repetitions is small, due to the economic analysis it may be interesting material for a potential reader.
The first important question that arises at the beginning of reading the material and methods chapter is whether the research was approved by the local ethics committee?
In sub-section 2.1, information on the health status of animals selected for the experiment should be completed.
In the material and methods section, there is no significant information about the stud male. Was the father one male or were there more? Information about breed, age etc. is essential and extremely important here. The "male" factor could significantly influence the results of the research.
In the first part of the discussion, the reader should once again be reminded of what questions (goals) the authors asked in their work. In this chapter, research hypotheses should also be marked and answered in such a way as to emphasize the achievement of the goal of the research undertaken.
L: 359-366 This paragraph lacks information on the effect of males used in reproductive on litter size and offspring weight.
Other minor comments:
L 168-173, 211-221 and 288-295 standard deviation cannot be "0", the results should be supplemented with a few consecutive decimal places.
L 321-323 this sentence should be included in the M&M section.
The Latin name for the species should be added. It seems that the name guinea pig is no longer used
Reviewer 2 Report
Unfortunately, this manuscript should not be accepted for publication in MDPI Animals.
I can only suggest the Authors to consult the appropriate scientific literature standards as in the following examples:
Davinia Sánchez-Macías, Noemí Castro, Miguel A. Rivero, Anastasio Argüello
& Antonio Morales-delaNuez (2016): Proposal for standard methods and procedure for guinea pig carcass evaluation, jointing and tissue separation, Journal of Applied Animal Research, 44:1, 65-70, DOI:10.1080/09712119.2015.1006234
Davinia Sánchez-Macías, Lida Barba-Maggia, Antonio Morales-delaNuez,
Julio Palmay-Paredes (2018): Guinea pig for meat production: A systematic review of factors affecting the production, carcass and meat quality, Meat Science, 143, 165-176, DOI:10.1016/j.meatsci.2018.05.004
Carlos Mínguez Balaguer, Antonio Calvo Capilla, Víctor Alfredo Zeas Delgado,
Davinia Sánchez Macías (2019): A comparison of the growth performance, carcass traits, and behavior of guinea pigs reared in wire cages and floor pens for meat production, Meat Science, 152, 38-40, DOI:10.1016/j.meatsci.2019.02.012
Author Response
Unfortunately, this manuscript should not be accepted for publication in MDPI Animals.
I can only suggest the Authors to consult the appropriate scientific literature standards as in the following examples:
Regards
Dear Reviewer, all authors are grateful for the time the review it. We have taken positive his suggestions and comment. These references have been a great input our work.
Reviewer 3 Report
Influence in litter size at birth on productive parameters in Guinea pigs
Dear Editors,
I read the submitted manuscript and after careful overall consideration the current status of the paper needs “minor revision” and then it´s suitable for publication.
The aim of the study was an analysis of the optimal litter size for the productive parameters in guinea pigs which are used for food production in South America (e.g. Peru, Bolivia, Ecuador and Colombia).
Different parameters were analyzed like final weight of the litter depending on the litter size. Furthermore the food conversion and the weight of the carcasses were used to find the optimal litter size for the best economic outcome.
What is missing for me is the discussion if there is a clue (age of the mother /father, weight of the mother or something else) how to get the optimal litter size of three pups? I think for the argumentation, that it might help the meat industry, it seems important for me to get an idea how it´s possible to get the wanted litter size of three or four pups? Because killing of the litter when the number of pups is to low or the surplus of the animals is not an option.
So in my opinion the aspect of the breeding management to get the “optimal” litter size is quite important.
Here some points in detail:
Introduction
Page 1
Line 39 and 44: fuse this two sentences about the precocity etc. it´s a little bit redundant
Line 62: Skip are
Material and Methods
Line 113: skip where Light cycle, time of injection?
Discussion
In the discussion the aspect how to get the optimal litter size in the meat production industry is missing.
Author Response
We have proceeded as recommended
Please see the attachment
